# Clinical Significance of Tumor Grade in Triple-Negative Breast Cancer: A Retrospective Cohort Analysis

**DOI:** 10.3390/biomedicines13051100

**Published:** 2025-05-01

**Authors:** Neya Ramanan, Mah-noor Malik, Sarang Upneja, Haniya Farooq, Swati Kulkarni, Rasna Gupta, John Mathews, Abdullah Nasser, Alina Bocicariu, Laurice Arayan, Lisa Porter, Bre-Anne Fifield, Rong Luo, Muriel Brackstone, Caroline Hamm

**Affiliations:** 1Schulich School of Medicine and Dentistry, Western University, Windsor, ON N8W 2X3, Canada; 2School of Medicine, University College Dublin, D04 V1W8 Dublin, Ireland; 3Department of Oncology, Windsor Regional Hospital, Windsor, ON N8W 2X3, Canada; 4Department of Pathology, Windsor Regional Hospital, Windsor, ON N8W 2X3, Canada; 5Department of Biomedical Sciences, University of Windsor, Windsor, ON N9B 3P4, Canada; 6Leddy Library, University of Windsor, Windsor, ON N9B 3P4, Canada; 7Department of Surgery, University of Western Ontario, London, ON N6A 5C1, Canada

**Keywords:** breast cancer, triple negative, grade, relapse, survival

## Abstract

Triple-negative breast cancer (TNBC) is a heterogeneous cancer that lacks estrogen receptors (ER), progesterone receptors (PR), and human epidermal growth factor receptor 2 (HER2) proteins. Here, we investigated the prognostic value of grade in patients with TNBC. **Methods:** This retrospective study analyzed 780 TNBC patients from two large regional cancer programs in Canada. Patients seen between 1 January 2004 and 31 December 2022 were included. Patients with grade 1 tumors and stage IV disease were excluded from analysis. Demographic information regarding the patient, tumor, and treatment were collected. The primary outcomes, relapse-free survival (RFS) and overall survival (OS), were analyzed using the Cox proportional hazards model and max-combo test. **Results:** For patients with grade 2 TNBC, median RFS was 14.1 years (95% CI, 9.48 to not reached (NR)) while it was not reached for patients with grade 3 tumors. No difference for relapse was identified in the first five years. Beyond 5 years, 4.9% of the patients with grade 2 tumors and 1.6% of those with grade 3 tumors relapsed (*p* = 0.006). In that same study period, 10.4% of patients with grade 2 tumors and 5.7% of those with grade 3 tumors died (*p* = 0.03). **Conclusion:** Grade 2 TNBC was associated with a higher risk of relapse and death after the 5-year mark compared to grade 3 TNBC. This distinct pattern of relapse and survival in grade 2 TNBC, characterized by an increased risk of relapse and mortality after 5 years, warrants confirmatory investigations.

## 1. Introduction

Triple-negative breast cancer (TNBC) is a subtype of breast cancer that lacks receptors for estrogen (ER) and progesterone (PR), as well as overexpression of human epidermal growth factor receptor 2 (HER2) proteins [1,2]. Clinical features more commonly seen in TNBC are a younger age at diagnosis, larger tumor size, higher histological grade, and earlier relapse [3,4,5]. The lack of ERs and PRs has limited the treatment of TNBC to chemotherapy initially, which has resulted in inferior outcomes for this patient group. Significant improvements in the treatment of TNBC continue to emerge and include immunotherapy, adjuvant and neo-adjuvant chemotherapies, and antibody–drug conjugates. The addition of platinum and check-point inhibitors in the neoadjuvant setting and capecitabine in the adjuvant setting have led to improvements in outcomes for TNBC [6,7,8].

Research by Perou into the molecular subtyping of triple-negative breast cancer suggested four molecular subtypes of breast cancer: ER+/luminal-like, basal-like, Erb-B2+, and normal breast. He also found that the clinical designation of estrogen receptor-negative breast carcinoma encompassed at least two distinct subtypes of tumors (basal-like and ErB-B2-positive), which he stated may need to be treated as distinct diseases [9]. Since then, with high throughput omics-based research, up to seven subtypes of TNBC have been reported: basal-like (1 and 2), immunomodulatory, mesenchymal, mesenchymal stem-like, and luminal androgen receptor (LAR) subtype of which the LAR subtype displayed patients with decreased relapse-free survival and histologically lower grades [10,11]. Omics have proven to be more difficult than expected to translate these findings into clinical use [12,13].

Histological tumor grade is an established prognostic factor of breast cancer and is factored into decisions on treatment for patients with breast cancer [14]. The American Joint Committee on Cancer (AJCC) updated to the eighth edition in 2018, incorporating biomarkers and grade into the staging system, effectively upstaging TNBC [15,16].

Previous research suggested a difference in outcomes based on grade for TNBC [17,18]. The purpose of this study was to validate those findings regarding the prognostic value of grade in TNBC by increasing the initial sample size and expanding to a second cancer program.

## 2. Materials and Methods

### 2.1. Study Participants

The TNBC database for this study consisted of 780 TNBC patients diagnosed between 1 January 2004 and 31 December 2022: 328 were from the Windsor Regional Cancer Program (WRCP) and 452 were from the London Health Sciences Centre (LHSC). TNBC was defined as ER less than or equal to 10%, PR less than or equal to 10%, and HER2 receptor 1+ or 2+ by IHC or negative by HER2 gene amplification when inconclusive. These patients were previously entered into institutional research databases. Research ethics approval was provided by the Windsor Regional Hospital Research Ethics Board #19193 and the Western University Research Ethics Board #119967.

### 2.2. Inclusion and Exclusion Criteria

TNBC was defined as less than or equal to 10% ER and PR expression, and low or zero HER2 expression by IHC or by HER2 gene amplification. The rationale for including up to 10% ER and/or PR positivity was based on current clinical practice and clinical trials [1,19,20,21]. Grade 1 tumors were excluded from analysis because of the small sample size, unique heterogeneity, and clinical outcomes of grade 1 TNBC [22,23,24]. Patients were excluded from this study if they were diagnosed with Ductal Carcinoma In Situ or stage 4 TNBC. Stage I, II, and III TNBC patients were included.

### 2.3. Demographics

The variables retrospectively collected from the database were demographics regarding the patient (age and BRCA 1/2 status), tumor (size, grade, ER/PR/HER2 status, pathology type, lymph node status, and cancer laterality), treatment (surgery, radiation, chemotherapy, and hormone therapy), and AJCC-7 stage.

Nineteen patients had grade 1 tumors and were excluded. The remaining 761 grade 2 and 3 patients were analyzed.

### 2.4. Outcomes

#### Statistical Analysis Methods for Relapse-Free and Overall Survival

A Cox proportional hazard model was employed to estimate the hazard ratios for relapse-free and overall survival between patients with grade 2 and grade 3 TNBC. Due to crossover in survival curves for these groups, the analysis was conducted by using a piecewise hazard ratio model, including cancer grade and time-period factors in the model along with the cancer-grade-by-time-period interaction term, allowing for a different hazard ratio in each period, with a *p*-value generated by a log-rank test for each interval. In addition, the max-combo test was conducted to test for relapses and overall survival differences. This analysis was based on an adaptive procedure involving selection of best test statistics with log-rank (G0,0) and the Fleming–Harrington test (G0,1, G1,0, and G1,1) with alpha correction for proportional hazards (PH) and non-proportional hazards (NPH). This provided a robust test under different scenarios including PH, delayed effect, crossing survival, early separation, and a mixture of more than one NPH–type scenarios. The association between age, grade, receipt of chemotherapy, and BRCA status was examined using a Wilcoxon rank sum test and Pearson’s Chi-squared test or Fisher exact test as appropriate.

## 3. Results

### 3.1. Demographics

A total of 780 TNBC patients were examined in this study. Nineteen patients (2.4%) with grade 1 tumors were removed, leaving 761 patients. A total of 127 (16.3%) patients were grade 2 and 634 (81.3%) were grade 3. Three patients were removed due to inadequate information on follow-up (Figure 1)

Almost one-third of patients presented with stage I tumors, and more than half had stage II tumors. Seven patients had inflammatory TNBC. Infiltrating ductal cancer not otherwise specified was the main histological pathology subtype with 645 patients (84.8%); infiltrating ductal with other features in 69 (9.1%), ‘other’ in 45 patients (5.9%), and missing in 2 (0.3%). ‘Other’ subtypes included squamous metaplasia, mucinous, papillary, cystic adenoid, or atypical medullary. See Table 1 for further patient characteristics.

### 3.2. Association Between Grade and Demographic Variables

The overall median age at diagnosis was 63 years but was older for those with grade 2 tumors with a median of 69 years of age (range 35–99 years of age) as compared to 62 years of age for those with grade 3 tumors (*p* < 0.00161). BRCA testing was available for 173 patients. Of these, 26.6% were positive for BRCA1 or BRCA2. There was no association between grade and BRCA positivity.

Grade 3 tumors demonstrated a statistically significant larger tumor size (*p* < 0.01). Grade 3 tumors were more likely to be ER zero (*p* = 0.026). There was no relationship between grade and PR status.

A larger proportion of patients with grade 3 tumors received chemotherapy in comparison to patients with grade 2 tumors, with 70.3% of grade 3 patients receiving chemotherapy and 57.9% of grade 2 patients receiving chemotherapy (*p* = 0.006). Patients with grade 2 tumors were more likely to receive endocrine therapy, with 7.1% of patients with grade 2 tumors and 3.2% of grade 3 patients receiving endocrine therapy (*p* = 0.035) (Table 2)

### 3.3. Grade and Relapse-Free Survival

For grade 2 patients, the median relapse-free survival (RFS) was 14.1 years (95% CI, 9.48 to not reached) while it was not reached for grade 3 patients (Figure 2). Within 0–5 years, 33 out of 123 grade 2 patients (26.8%) compared with 145 out of 628 grade 3 patients (23.1%) experienced relapse. The hazard ratio was 0.902 (95% CI: 0.617, 1.319) with a *p*-value of 0.600. This suggested that there was no significant difference in RFS between grade 2 and grade 3 patients during this time interval as indicated by the non-significant *p*-value.

Beyond 5 years, 6 out of 123 grade 2 patients (4.9%) compared with 10 out of 628 patients (1.6%) experienced relapse. The hazard ratio was 3.730 (95% CI: 1.351, 10.300) with a *p*-value of 0.006. This indicated a significantly higher risk of relapse in grade 2 patients compared to grade 3 patients after 5 years, which suggested that grade 3 patients had more favorable long-term outcomes.

The max-combo two-sided adjusted *p*-value was 0.015, which indicated overall significant differences in RFS between each grade when considering multiple testing. Fleming–Harrington weighted log-rank test results further illustrated the variability in survival distributions, with the FH(1,1) and FH(0,1) tests indicating significant differences between groups, particularly in mid- to longer-term survival scenarios.

### 3.4. Grade and Overall Survival

The median overall survival (OS) was 15.3 years (95% CI, 14.10 to not reached) for grade 2 patients and not reached (95% CI, 8.70 to not reached) for grade 3 patients (Figure 3). In the 0-to-5-year interval, 28 out of 125 grade 2 patients (22.4%) compared with 145 out of 628 grade 3 patients (23.1%) died. The hazard ratio was 0.753 (95% CI: 0.496, 1.143) with a *p*-value of 0.200. This suggested that there was no significant difference in OS between grade 2 and grade 3 patients during this time interval.

For the greater-than-5-year study period, 13 out of 125 grade 2 patients (10.4%) compared with 36 out of 628 grade 3 patients (5.7%) died. The hazard ratio was 2.029 (95% CI: 1.076, 3.827) with a *p*-value of 0.030, indicating a significantly higher risk of death in grade 2 patients compared to grade 3 patients after 5 years.

The max-combo two-sided adjusted *p*-value was 0.068, indicating a non-overall survival significant difference when multiple testing was considered. Fleming–Harrington weighted log-rank test results illustrated the variability in survival distributions, with the FH(1,1) and FH(0,1) tests indicating significant differences between groups, particularly in middle- and longer-term survival scenarios.

### 3.5. Relapse-Free and Overall Survival Hazard Ratios

It can be seen in both Kaplan–Meier curves for relapse-free and overall survival (Figure 2 and Figure 3) after 5 years from time of diagnosis that grade 2 patients that had initially fared better than grade 3 patients experienced inferior outcomes. A 5-year cut point was conservatively chosen to stratify these patients as that was when the curves can be seen to separate from each other. Grade 2 patients had a 3.7-fold inferior time to relapse after 5 years from the time of diagnosis (HR 3.730; 95% CI 1.35–10.1; *p* = 0.006). Furthermore, grade 2 patients were shown to have a 2.0-fold increased risk of death after the first five years from the time of diagnosis (HR 2.029; 95% CI 1.08–3.83; *p* = 0.03). Neither of the differences were statistically significant prior to five years.

Adjusting for age and rate of receipt of chemotherapy, for RFS, the HR for the <5-year range was 0.82 (95% CI: 0.56–1.21) and for the >5-year range was 2.96 (95% CI: 1.04–8.47). For OS, the HR for the <5-year period was 0.690 (95% CI: 0.46–1.03), and for the >5-year range, the HR was 1.53 (95% CI: 0.79–2.96).

## 4. Discussion

Unique in this study was the long-term follow-up, not normally reported in prospective studies, which allowed the identification of a different pattern of relapse and death between grade 2 and grade 3 TNBC patients. Although no difference in risk of relapse or death was identified in the first five years of follow-up, after five years we identified a statistically significantly higher risk of relapse and death in patients that had grade 2 tumors than patients that had grade 3 tumors. The overall max-combo analysis and specific weighted log-rank tests reinforced the conclusion that grade 3 patients had more favorable long-term outcomes for both relapse-free and overall survival compared to grade 2 patients.

It has been reported that patients with TNBC have a low risk of relapse after 5 years [1,10,25]. We find this report consistent with our grade 3 TNBC; however, the grade 2 TNBC relapse curve displayed an ongoing risk for relapse after 5 years not previously reported. In previous research by Upneja et al., 305 TNBC patients from WRCP were assessed for RFS and OS. Grade 2 patients were found to have significant RFS and OS when compared to grade 3 [18]. We expanded the database to include more patients from both WRCP and LHSC to increase generalizability and to assess the significance further, and we determined that the significance for relapse-free survival persists regardless of increased sample size.

Johansson reported the outcomes of 1911 TNBC patients in the Norwegian Registry treated between 2005 and 2015. This study reported a similar pattern of relapse that was identified in our study, with grade 3 having an initial higher risk, and grade 2 showing a higher risk of relapse after 10 years [26]. They reported grade 3 patients having a higher risk of relapse in the first 5 years; however, their curves crossed 10 years later than the 5 years in our study. In their study, they only had systemic treatment data on 57% of patients, similar to our findings [26].

BRCA status did not impact outcomes, which was similar to the findings of Park et al. [27]. We also identified no relationship between grade and BRCA positivity.

The LAR subtype was characterized as having biologically more aggressive tumors with frequent metastases in regional lymph nodes and the lowest rate of pathological complete response to chemotherapy. The LAR subtype showed a lower histologic grade than the non-LAR subtype with statistically significantly more in this category having grade 2 tumors and fewer with grade 3 tumors [28]. In the same study, they identified that the LAR subgroup had the oldest age, similar to our grade 2 population who were older than the grade 3 population.

In one study, clinical outcomes were examined using the LAR subtypes. The 5-year distant disease-free survival (DDFS) rates were 67.2% for LAR+ and 80.6% for LAR- patients (HR = 1.82 95%CI 1.10–3.02, *p* = 0.020), showing LAR+ tumors had a worse DDFS when compared to LAR- in TNBC patients [29]. The LAR+ subgroup did have an earlier pattern of relapse than the grade 2 patients with higher rates of relapse occurring after 2 years.

### 4.1. Limitations of Our Research

There are inherent limitations of a retrospective chart review. In addition, almost one-third of patients in this data set did not receive chemotherapy. Statistically more patients with grade 3 cancer received chemotherapy: 42.1% of grade 2 patients did not receive chemotherapy compared to only 29.7% of grade 3 patients (*p* = 0.006). Immunotherapy has become the standard of care for stage II and III TNBC [6]. Adjuvant capecitabine has been shown to effectively prolong relapse-free and overall survival in triple-negative breast cancer patients [6,7]. Immunotherapy was approved for use in this patient population in June 2022, and so no patients in this study received immunotherapy. Only five patients received capecitabine and none received immunotherapy, so it is not known that these findings would be realized with today’s current standard of care. Future efforts to assess the impact of grade in a prospective clinical trial would confirm the findings of this research.

We were not able to collect breast cancer-specific survival, which would have strengthened our survival results.

### 4.2. Recommendations

Validation of these findings is needed to determine if in fact tumor grade in triple-negative breast cancer is an independent predictor of long-term relapse-free and overall survival. The effect of newer treatments on the pattern of relapse needs to be examined. KEYNOTE-522 reporting the benefits of immunotherapy in TNBC did not stratify their results by grade, and outcomes after five years would have to be analyzed [6]. The CREATE-X study combined grade 2 and 3 in their analysis and longer-term follow-up would need to be analyzed [7].

## 5. Conclusions

In this multicenter retrospective study, we determined that TNBC patients with grade 2 tumors were shown to have significantly inferior relapse-free survival than grade 3 patients, with poorer relapse-free and overall survival after 5 years from the time of diagnosis. Further research into this finding in the era of immunotherapy and adjuvant capecitabine is needed to validate these findings using today’s landscape of treatments.

## Figures and Tables

**Figure 1 biomedicines-13-01100-f001:**
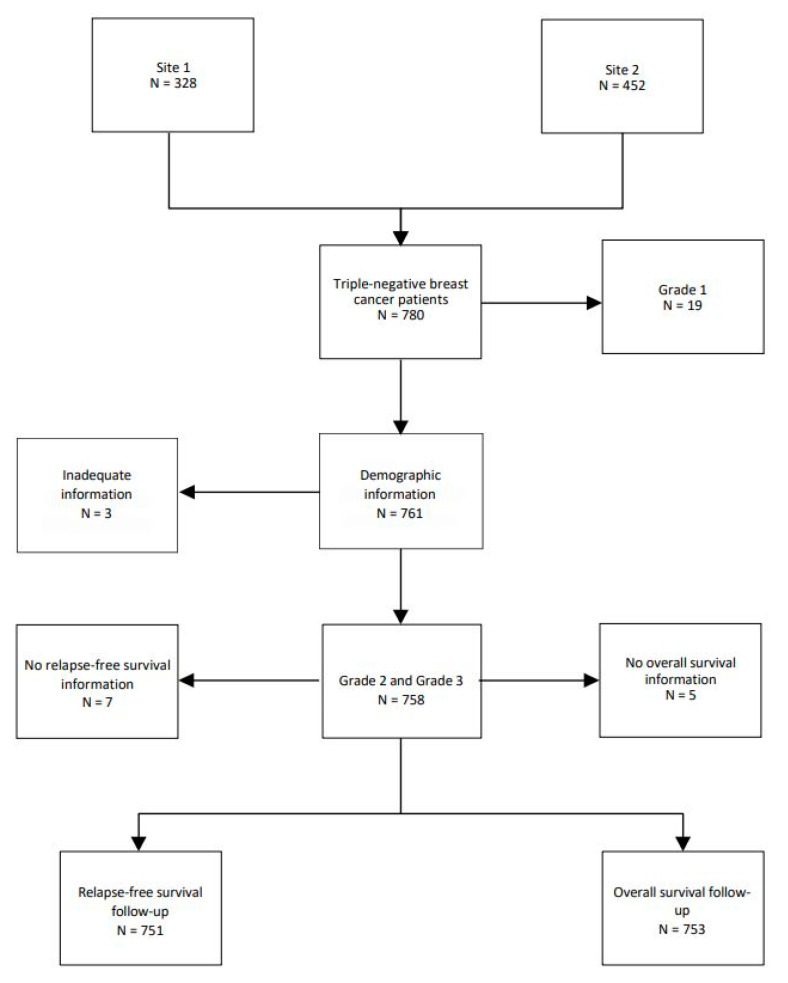
Flow chart of TNBC patients.

**Figure 2 biomedicines-13-01100-f002:**
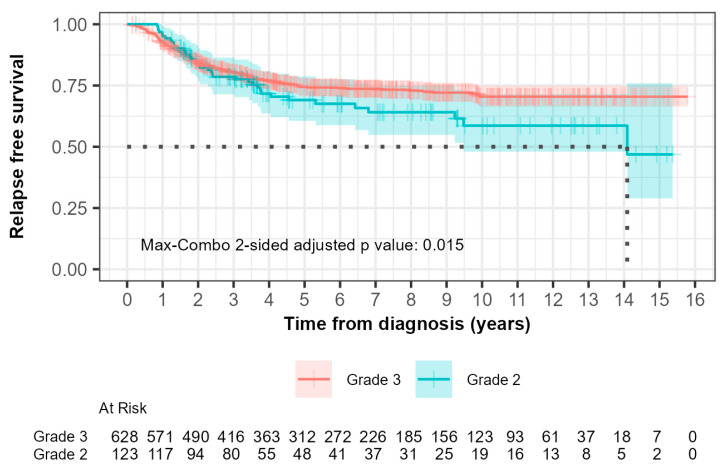
Kaplan–Meier curves of relapse-free survival (RFS) comparing patients with grade 2 and grade 3 tumors. The median RFS times, demonstrated by the black dotted line, were 14.1 years and not reached for grades 2 and 3, respectively. (*p* = 0.01).

**Figure 3 biomedicines-13-01100-f003:**
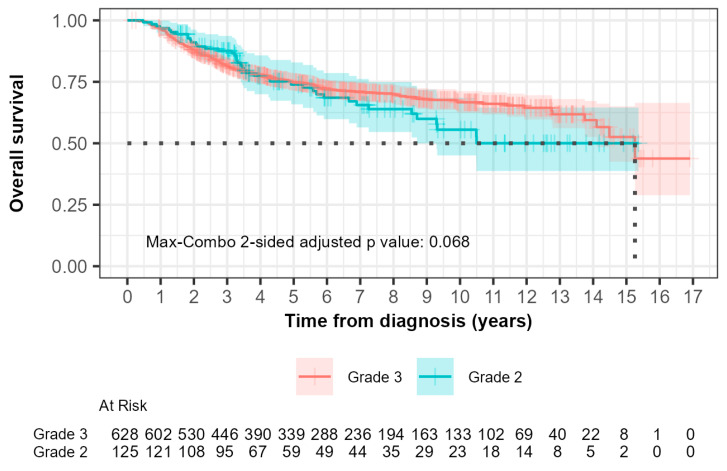
Kaplan–Meier curves of overall survival (OS) comparing patients with grade 2 and grade 3 tumors. The median OS, demonstrated by the black dotted line was 15.3 years (95% CI, 14.10 to not reached) for grade 2 patients and not reached (95% CI, 8.70 to not reached) for grade 3 patients.

**Table 1 biomedicines-13-01100-t001:** Demography, BRCA status, and management options for TNBC patients.

Characteristic		*n* (%)
Median age at diagnosis (years)	63 (25–99)	
Cancer stages	Stage I	238 (31.3%)
	Stage II	396 (52.0%)
	Stage III	126 (16.6%)
	Missing information	1 (0.1%)
Type of chemotherapy	Anthracycline based	71 (9.3%)
	Taxane based	41 (5.3%)
	Taxane + Anthracycline-based	268 (35.2%)
	Taxane/Anthracycline/Carboplatin-based	113 (14.5%)
	CMF *	9 (1.2%)
	No chemotherapy	247 (32.5%)
	Other **	12 (1.5%)
Type of surgery	Mastectomy	37 (4.9%)
	Mastectomy and ALND ^¥^	73 (22.7%)
	Mastectomy and SLNB ^§^	103 (13.5%)
	Lumpectomy	18 (2.4%)
	Lumpectomy and ALNB	115 (15.1%)
	Lumpectomy and SLNB	297 (39.0%)
	None	11 (0.1%)
	Other	16 (2.1%)
	Missing information	11(0.1%)
BRCA status	BRCA1 positive	77(15.6%)
	BRCA2 positive	19 (11.0%)
	BRCA unknown	715 (73.4%)

***** CMF: cyclophosphamide, methotrexate, 5-fluorouracil; ** other included trastuzumab, gemcitabine, and capecitabine; ^¥^ ALND: axillary lymph node dissection; ^§^ sentinel lymph node biopsy.

**Table 2 biomedicines-13-01100-t002:** Association of demographic and clinical variables with tumor grades.

	Grade 2 (*n* = 127)	Grade 3 (*n* = 634)	*p*-Value
Age at diagnosis			< 0.001 ^1^
Median	69.0	62.0	
Range	35.0–99.0	25.0–105.0	
Size (cm)			< 0.001 ^1^
Median	2.0	2.5	
Range	0.1–15	0.1–15	
Missing information	2	3	
No. of Lymph nodes positive			0.903 ^1^
Median	0	0	
Range	0–21.0	0–35.0	
Missing information	5	16	
Estrogen receptor status %			0.021 ^2^
0%	119 (93.7%)	618 (97.5%)	0.026 ^2^
1–9%	7 (5.5%)	16 (2.5%)	0.073 ^2^
=10%	1 (0.8%)	0 (0%)	0.025 ^2^
Progesterone receptor status %			1.000 ^2^
0%	123 (96.9%)	609 (96.1%)	0.670 ^2^
1–9%	4 (3.2%)	24 (3.8%)	0.728 ^2^
=10%	0 (0%)	1 (0.1%)	0.654 ^2^
Cancer stages			
Stage 1	51 (40.2%)	187 (29.5%)	
Stage 2	56 (44.1%)	340 (53.7%)	
Stage 3	20 (15.8%)	106 (16.8%)	
Missing information	0	1	
Chemo regimen received			0.006 ^2^
Yes	73.0 (57.9%)	441.0 (70.3%)	
No	53.0 (42.1%)	186.0 (29.7%)	
N-Miss	1	7	
Radiation site			0.938 ^2^
N-Miss	1	1	
No	37 (29.4%)	171 (27.0%)	
Breast	66 (52.4%)	334 (52.8%)	
Chest Wall	22 (17.5%)	121 (19.1%)	
Both	1 (0.8%)	7 (1.1%)	
Hormone therapy			0.035 ^2^
Yes	9 (7.1%)	20 (3.2%)	
No	118 (92.9%)	614 (96.9%)	

^1^. Wilcoxon rank sum test; ^2^. Pearson’s Chi-squared test or Fisher exact test as appropriate.

## Data Availability

Data are available at https://borealisdata.ca/dataverse/westernu (accessed on 2 October 2024).

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
