# Peer review of "Clinical Significance of Tumor Grade in Triple-Negative Breast Cancer: A Retrospective Cohort Analysis"

_biomedicines, 2025, doi:10.3390/biomedicines13051100_

Round 1
Reviewer 1 Report
Comments and Suggestions for Authors
Thank you very much for the manuscript. Please see below some inputs and suggestions.
Title
- Add ‘Tumor’ to make it ‘Tumor Grade’.
Abstract
- Delete “tumor” in Line 26. “Demographic information regarding the patient, tumor, cancer, and treatment..” Tumor and cancer appear to have the same meaning, unless if referring to tumor microenvironment.
- Otherwise, the abstract is fine.
Keywords
- Consider adding outcomes or prognosis in general or specific outcome i.e. relapse and survival
- Otherwise, no other concerns.
Materials and Methods: No concerns.
Data Analysis
- Must include how the influence age, BRCA and treatment options on relapse free and overall survival were tested (e.g. logistic regression).
- Comments on their influence are included in the results (Results: not statistically significant) and should therefore have been mentioned in the plan for data analysis.
Results
- Figure 1: The legend for figures must be placed at the bottom.
- Table 1: The legend has to be comprehensive and self-explanatory. “Demographics” is vague. Furthermore, the table not only contain demographic information. Consider ‘Demography, BRCA status and management options for patients with TNBC.
- Table 2
- Write 0 as just 0 instead of “0.0”.
- Figure 2: Place the legend immediately below the figure and not before, although there is a second ligand.
- Figure 3: Place the legend at the bottom.
- Report 95% CI consistently. For example: (95% CI: 1.076,827) versus (95% CI: 0.46-1.03).
Discussion
- Some of the paragraphs are short. They may be merged without affecting the flow.

Reviewer 2 Report
Comments and Suggestions for Authors
Dear Authors,
Kindly include the suggested changes to make the manuscript more informative.
The submitted manuscript may be accepted for publication, provided the authors agree to address the suggested modifications.
- Review Comments on the Manuscript: “Clinical Significance of Grade in Triple Negative Breast Cancer: A Retrospective Cohort Analysis” by Ramanan et al.
The submitted manuscript presents a commendable effort to evaluate the correlation between tumor grade and survival outcomes in patients with triple-negative breast cancer (TNBC) undergoing different anticancer treatment modalities. The study addresses a clinically relevant question and provides valuable insights that may assist clinicians and researchers in optimizing treatment selection based on prognostic indicators such as tumor grade.
The manuscript is well written and structured; however, the Introduction section requires significant enhancement. It is currently too brief and lacks essential background information on several key areas such as,
- A concise overview of triple-negative breast cancer, including its clinical and molecular characteristics.
- The role and challenges associated with treating TNBC, particularly in the absence of hormone receptors and HER2 expression.
- A brief summary of current anticancer treatment regimens commonly used for TNBC.
- The prognostic and therapeutic implications of estrogen and progesterone receptors in breast cancer subtypes, to provide context for why TNBC presents a unique therapeutic challenge.
Furthermore, while the authors have acknowledged the limitations of their study, it would be beneficial to include potential strategies to address these limitations in future research.
By elaborating on these points, the manuscript would not only demonstrate critical self-assessment but also offer direction for future investigations, thereby increasing its impact.
Best Regards,
Reviewer 3 Report
Comments and Suggestions for Authors
Dear Authors,
the manuscript “Clinical Significance of Grade in Triple Negative Breast Cancer: A 2 retrospective cohort analysis“ by Neya Ramanan1,*; Mah-noor Malik1,*; Sarang Upneja, MD1; Haniya Farooq2; Swati Kulkarni, MD1,3; Rasna Gupta, MD1,3; 4 John Mathews, MD1,3; Abdullah Nasser, MD1,3; Alina Bocicariu, MD 1,4 Laurice Arayan1; Lisa Porter, PhD5; Bre-Anne 5 Fifield, PhD5; Rong Luo, PhD6; Muriel Brackstone, MD1,7; Caroline Hamm, MD1,3,** has been revised.
In their manuscript, the authors describe the investigation of the prognostic value of grade in patients with TNBC. This retrospective study analyzes 780 TNBC patients from two large regional cancer programs in Canada. Patient’s seen between January 1, 2004 and December 31, 2022 were included. Patients with grade 1 tumors and 24 stage IV disease were excluded from analysis. Demographic information regarding the patient, tumor, cancer, and treatment were collected. The primary outcomes, relapse-free survival and overall survival, were analyzed using the Cox proportional hazards model and the max-combo test. The authors determined that TNBC patients with grade 2 tumors were shown to have significantly inferior relapse-free survival than grade 3 patients, with poorer relapse-free and overall survival after 5 years from the time of diagnosis.
The study could make a positive contribution to the relevant scientific field. In order for the manuscript to be published, authors should improve it by making the following corrections/additions:
-The Introduction section should be supplemented with the information about TNBC, existing and most frequently used treatment methods, and tumor grades and staging system as well.
-The impact of the type of cancer treatment (chemotherapy, surgery, hormone therapy) on recurrence and survival should be discussed.
-L81: ...Figure 3), ….
-L98: Figure 1 should be reduced. There is no need to upload such a large image.
-L103-107: Which part of Table 1 shows the presented information. Please, discuss the data in Table 1 clearly.
-In Table 1: Stages 1, 2, 3 should be changed with Stages I, II, III. CMF should have * (an asterisk).
-L117: according to Table 2, the range 25-99 should be improved to 35-99.
L130: In Table 2: Please move “Grade 2” above “(N=127)”.
-L132, 133: Please move these lines closer to the table.
-L136: … relapse free survival (RFS).....
-Throughout the text, please standardize the spelling of “relapse free survival“ (or relapse-free survival).
-L161: ...overall survival (OV)....
-L183: …(Figures 2 and 3)….
-In the Discussion section: A broader comparison of the manuscript data and Ref [13] data should be provided.
Round 2
Reviewer 3 Report
Comments and Suggestions for Authors
Dear Authors,
the manuscript "Clinical Significance of Grade in Triple Negative Breast Cancer: A retrospective cohort analysis" by Authors Neya Ramanan , Mah-noor Malik , Sarang Upneja , Haniya Farooq , Swati Kulkarni , Rasna Gupta , John Mathews , Abdullah Nasser , Alina Bocicariu , Laurice Arayan , Lisa Porter , Bre-Ann Fifield , Rong Luo , Muriel Brackstone , Caroline Hamm * (manuscript ID: biomedicines-3592141) is properly improved and now is suitable for publication in the journal "Biomedicine".